# A Comparative Study on the Effect of Euthanasia Methods and Sample Storage Conditions on RNA Yield and Quality in Porcine Tissues

**DOI:** 10.3390/ani13040698

**Published:** 2023-02-16

**Authors:** Bimal Chakkingal Bhaskaran, Roel Meyermans, Wim Gorssen, Gregory Erich Maes, Steven Janssens, Nadine Buys

**Affiliations:** 1Center for Animal Breeding and Genetics, Department of Biosystems, KU Leuven, Kasteelpark Arenberg 30, Box 2472, 3001 Leuven, Belgium; 2Center for Human Genetics, Genomics Core, UZ-KU Leuven, 3000 Leuven, Belgium

**Keywords:** euthanasia method, nitrogen foam anoxia, T-61^®^, RNA*later*™, liquid nitrogen, RNA yield and quality, pigs

## Abstract

**Simple Summary:**

Animals have been used as subjects in biomedical research for a long time. In these studies, tissue collection for RNA profiling is often essential, and hence, the animals are euthanised using standard injectable anaesthetics such as sodium pentobarbital or T-61^®^. Recently, a euthanasia method using an inhalant anaesthetic ‘nitrogen gas in foam’ (ANOXIA^TM^) has gained further interest as it is claimed to be more animal-friendly. However, little is known on its effect on subsequent RNA analysis. There are also no studies on the interaction effect of these euthanasia methods and different tissue storage conditions such as RNA*later*™ or snap freezing using liquid nitrogen on RNA measurements. It is very important to investigate if the choice of euthanasia method in research animals as well as the tissue storage condition impact RNA measurements. The current study compared the two euthanasia methods in male piglets and found that the nitrogen anoxia technique (ANOXIA^TM^) could be a suitable alternative to T-61^®^ based on RNA quality parameters because no differences were detected. Storage in RNA*later*™ significantly increased the RNA integrity in comparison to snap freezing, but no interaction effect of the euthanasia methods and storage conditions on RNA measurements were detected.

**Abstract:**

Animals used in research often have to be euthanised, especially when tissue sampling is essential. Recently, a euthanasia method, utilizing an inhalant anaesthetic ‘nitrogen gas in foam’ in an anoxia box (ANOXIA^TM^), has gained considerable interest as it claimed to be more animal-friendly. However, it is not clear whether the use of this euthanasia method has an influence on RNA measurements. Moreover, there are no studies on the interaction effect of different euthanasia methods on the tissue sample storage conditions. The current study compared RNA measurements from two euthanasia methods (ANOXIA^TM^ vs. T-61^®^ injection) and two storage conditions (RNA*later*™ vs. snap freezing) in 12 male piglets. The nitrogen anoxia method had a significantly higher RNA yield (*p* < 0.01) compared to the T-61^®^ method. However, no effect of the euthanasia methods on the A260/230 ratio and RIN value was observed. Tissues stored in RNA*later*™ had significantly higher RIN values (*p* < 0.001) compared to snap frozen samples. The present study could not find a significant difference between the two euthanasia methods in piglets, with regard to RNA quality measurements. Hence, the nitrogen anoxia technique (ANOXIA^TM^) might be considered as a suitable alternative to T-61^®^ for euthanasia of piglets used in research.

## 1. Introduction

Animals used in experiments for scientific purposes are often euthanised, for example, to harvest tissues for further testing. Choice of a euthanasia method depends on its suitability with subsequent research as well as its compliance with animal welfare standards. In its recommendations on euthanasia of experimental animals, the European Commission advises that the approved methods for euthanasia must ensure humane killing and consider welfare aspects of animals [1]. The American Veterinary Medical Association (AVMA) emphasises the need for careful consideration of the decision to euthanise, as stated in its guidelines on euthanasia of animals [2]. The AVMA recognises the practical necessity of this killing but stipulates that the act should be carried out while adhering to strict policies, guidelines and applicable regulations. As per the European Council Regulation 1099/2099 [3] and recommendations for euthanasia of experimental animals [4], approved methods for euthanasia of pigs used in research include use of injectable anaesthetics such as sodium pentobarbital (a barbiturate drug) and T-61^®^ (a nonbarbiturate compound drug marketed by Intervet Int via MSD AH) as well as inhalant anaesthetics such as argon and nitrogen (N_2_). Exposure to N_2_ gas in a stable anoxic atmosphere is considered to be a suitable alternative method for euthanasia in pigs [5,6]. A recent study investigated the feasibility of high concentration (98%) N_2_ gas stunning in pigs and its additional effect thereafter on meat quality traits [7]. The study concluded that this stunning method did not impose any adverse effect on meat quality and was recommended as a suitable alternative to electric as well as carbon dioxide (CO_2_) stunning. In another study, a euthanasia method utilizing high-expansion foam filled with N_2_ gas was tested in pigs, and the results pointed towards improved animal welfare as compared to high-concentration CO_2_ stunning [6]. It was reported that the animals did not show any aversion towards N_2_ during the initial exposure period and exhibited relatively less aversion to N_2_ and foam during the entire course of the euthanasia procedure, as compared to CO_2_ stunning. These results are promising on an animal welfare point of view, and hence, it would be highly valuable to test the suitability of this method for use in experimental animals such as pigs. However, the effect of N_2_ anoxia at the tissue level is still unclear, especially when samples are harvested for ribonucleic acid (RNA)-based studies.

RNA-based gene expression studies require efficient tissue handling techniques, primarily due to the highly fragile nature of the RNAs [8,9,10,11]. The purity and integrity of RNA have to comply with the requirements for downstream applications [10,11,12,13]. Often it is not feasible to immediately process the freshly sampled tissues, especially in field studies. Hence, these samples are flash frozen by immersion in liquid nitrogen (LN_2_) or by placing on dry ice and stored subsequently at −80 °C until further processing [14]. However, these tissue stabilisation methods are not always a viable option due to several limiting conditions. Firstly, the shipping and storage of LN_2_ presents a practical challenge in field conditions. Secondly, the rapid RNA degradation that occurs in unstabilised tissues while the frozen samples are thawed prior to RNA extraction is posing a challenge. A widely accepted alternative is to immerse the sample in RNA*later*™ (Invitrogen, Vilnius, Lithuania), a commercially available stabilising solution based on quaternary ammonium salts that permeates into the tissue and preserves the cellular RNA [15,16].

RNA has a maximum absorption at a wavelength of 260 nm, and the ratio of the measurements at 260 nm, 280 nm and 230 nm guides researchers to assess the purity of nucleic acids [17,18]. A 260/280 absorbance ratio (A260/280) of 2.0–2.1 is generally indicative of pure RNA, and a lower ratio indicates protein contamination [18]. Similarly, an ideal A260/230 ratio is higher than the respective A260/280 values and should range between 2.0 and 2.2, with lower levels indicative of contamination with guanidine salts or phenol [18]. Although there is no consensus on an acceptable lower limit for the A260/230 ratio in downstream analysis, it is important to take note of the respective RNA concentrations while evaluating the impact of lower ratios. If the RNA concentration is high, trace levels of contaminants will not have an impact on the absorbance ratios. However, these measurements become less reliable when the RNA concentrations are too low, as it could be an overestimate due to the contaminants present in the sample [17]. It is important to accurately assess the quantity and purity of RNA before proceeding. Another measure of paramount importance is the RNA Integrity Number (RIN), as it reflects the quality of RNA [19]. The RIN is measured using a microfluidic chip in a Bioanalyzer system (developed by Agilent Technologies). Here, an algorithm predicts and classifies the RNA integrity on a scale from 1 to 10, with the maximum value assigned for the most intact RNA [19,20,21]. A RIN value of 7 or above is considered technically essential for high-end downstream analysis such as qRT-PCR [12] and RNAseq [22].

The effect of various tissue storage conditions on the purity and integrity of the extracted RNA has been reported [14,23,24,25]. Tissue samples preserved in RNA*later*™ significantly improved RNA yield and integrity as compared with snap freezing [24]. Similarly, efficacy of RNA*later*™ in stabilizing RNA from blood samples was tested, and the results showed superior RNA yield and integrity [23]. Long-term effects of tissue storage on RNA yield and integrity under conditions such as freezing at −80 °C or vapour-phase liquid nitrogen (VPLN~−150 °C) were also previously tested [14]. RNA yield and integrity were reported to be significantly better for tissues that were stored at −80 °C compared to the ones stored in VPLN. Furthermore, other research studied the effects of storage conditions on gene expression profiles. Storage of tissues in RNA*later*™ for 24 or 72 h did not show any shift in quantitative gene expression when compared to fresh or frozen tissues [25]. Moreover, expression levels of gene signatures associated with breast cancer were found to be unaffected by the storage condition, whether stored in RNA*later*™ or snap frozen [24]. 

Impact of euthanasia methods on metabolomics of different mammalian tissues has been previously studied [26,27]. However, an aspect that is not studied is the effect of euthanasia methods on RNA yield and quality parameters as well as the possible interaction effect of the euthanasia method and storage condition on RNA features. In this context, this study aimed to test the ‘nitrogen in foam anoxia method’ using an anoxia box (ANOXIA^TM^) and compare it against the ‘T-61^®^ injection’ method, two of the approved methods for euthanizing piglets used in research. The mode of action involved in the N_2_ anoxia method is the hypoxia attained as a result of the rapid displacement of oxygen available to the animal [5]. T-61^®^ acts by inducing respiratory depression and muscular paralysis resulting in euthanasia of the animal [28]. In a controlled trial, we tested whether the nitrogen anoxia method would be a suitable alternative for euthanizing piglets used in gene expression studies. To evaluate these euthanasia methods, we compared RNA yield and purity (A260/280 and A260/230 ratio) as well as quality (measured as RNA integrity number-RIN value) of RNA extracted from several tissues. Moreover, it was evaluated whether the timing of the sampling as well as the combined effect of the euthanasia method and sample storage conditions had an influence on the RNA yield, purity and RNA quality parameters.

## 2. Material and Methods 

### 2.1. Experimental Set Up

Samples were collected from one-week-old purebred Pietrain male piglets (*n* = 12) from two litters (Litter I, *n* = 7; Litter II, *n* = 5). Within litters, the piglets were randomly assigned to one of the two methods of euthanasia. Only male piglets were used in this study as sampling was also planned to serve as a pilot experiment on cryptorchidism in one-week-old piglets. The experiment was conducted in accordance with the guidelines and approval of the Ethical Committee Animal Experimentation (ECD), KU Leuven [Ref No: P198/2018].

### 2.2. Euthanasia Method

Six animals from the experimental group were euthanised using an anoxia box- ANOXIA^TM^ (Anoxia, Putten, The Netherlands), where nitrogen gas in foam was used as the euthanizing agent. The manufacturer’s guidelines were followed for operating the anoxia box—ANOXIA^TM^ [29]. Six animals were euthanised using T-61^®^ injection (Intervet, via MSD AH). T-61^®^ is a combination drug, constituting the ingredients embutramide, mebozonium iodide, and tetracaine hydrochloride. Embutramide induces narcosis and respiratory depression, while mebozonium causes nondepolarizing muscular paralysis. Tetracaine is a local anaesthetic. This euthanasia method involved a primary sedation step using xylazine–ketamine injection (dose: 10 mg/kg intramuscular (i.m) injection of 10% ketamine + 2 mg/kg i.m injection of 2% xylazine), followed by T-61^®^ injection (Dose: 0.3 mL/kg intracardiac). Death was confirmed by auscultation of the heart and also by verifying absence of vital reflexes.

### 2.3. Sampling and Storage 

Tissue sections from the hypothalamus, pituitary, heart, lungs, liver, kidney, inguinal rings–superficial (SIR) and deep (DIR), cremaster muscle (CM) and testis were collected immediately after euthanasia. SIR, DIR and CM were collected as a part of the pilot experiment conducted prior to a subsequent study on cryptorchidism. Each tissue sample, except from the pituitary, was split into two (size less than 0.5 cm in thickness), and the split samples were stored under two different conditions, (i) in RNA*later*^TM^ and stored initially at 4 °C for 24 h and subsequently at −20 °C or (ii) initially snap frozen in liquid nitrogen (LN_2_) and thereafter stored at −80 °C, until RNA extraction. Pituitary samples were collected as soon as possible and were stored exclusively in RNA*later*^TM^ because of the small tissue size. The hypothalamus, DIR and SIR were collected, split into two and stored in both RNA*later*^TM^ and LN_2_. Heart, lungs, liver, testis, CM and kidneys were sampled on two time points to assess the influence of sampling time on RNA yield and purity. This was to simulate sampling under field conditions and to assess the decay of RNA over time. Care was taken to avoid the puncture site while sampling heart tissues from the T-61^®^ group, but there could be a bias introduced due to the intracardiac administration of the anaesthetic. Samples were collected and stored (i) as soon as possible after collection, within 30 min after euthanasia (early) and (ii) with a delay of 60 min after euthanasia (late). These samples were also split and stored under both conditions. Time elapsed from the start of the euthanasia procedure to the moment of death confirmation and the point of sample storage was recorded (in minutes) for each sample. Time of sampling was defined as the time difference (in minutes) between the point of sample storage and confirmation of death. Samples collected from each tissue type were thus grouped based on the euthanasia method, type of storage condition and the time of sampling. A total of 708 samples, comprising the ten different tissue types, were sampled and stored for RNA extraction. A detailed description of the sampling scheme is given in the Appendix A Appendix A.

### 2.4. RNA Extraction

For each tissue type, a sample with approximate dimensions of a 3 mm cube and weighing <30 mg was sliced off and used as the starting material. The samples were homogenised using Precellys^®^ Evolution tissue homogeniser (Bertin Technologies, Montigny le-Bretonneux, France). RNA was extracted using the Qiagen RNAeasy Mini Kit (Qiagen, Hilden, Germany) and was eluted in a volume of 50 µL RNAse-free water. A measurement of RNA concentration (ng/µL) could be used as an indirect estimate of total RNA extracted, as they are directly correlated, and RNA samples used in this experiment were eluted in a similar volume. The RNA concentration (ng/µL) and purity, measured as A260/230 ratio and A260/280 ratio, were estimated using a SimpliNano^TM^ Spectrophotometer (Biochrom, Cambridge, UK). The samples were selected randomly for RNA extraction, and RNA measurements from 489 samples were eventually available for further analysis. RIN values were measured for a subset of samples (*n* = 56) representing the tissue samples of pituitary, hypothalamus, lungs and liver (Appendix A Appendix A) to test the suitability for transcriptome analysis. Sampling from brain and lung also gives an additional opportunity later to examine for any possible effect of the euthanasia methods on these tissues at gene expression level. RIN value was estimated using the Agilent 2100 Bioanalyzer (Agilent Technologies, Diagem, Belgium).

### 2.5. Statistical Analysis

A filtering of the dataset was performed initially, based on the assumption that samples with low RNA concentration and low A260/230 ratio have unreliable quality measurements. Thus, a threshold of 50 ng/µL and 1.0 was set for RNA concentration and A260/230 ratio, respectively, and all the samples below the threshold were not retained for further analysis.

A linear model was used to evaluate the relationship between the dependent variables, RNA concentration; RNA purity, A260/280 and A260/230 ratios; and RIN values, and the predictor variables—euthanasia method, storage condition and tissue types. Time of sampling (recorded in minutes) was included as a covariate in the models for evaluating RNA purity and integrity while time of sampling and tissue weights (in milligrams) were included as the covariates in the model for RNA concentration. However, tissue weights were only available for 271 samples that were used in this analysis. The models were also tested by including the ‘time of sampling’ as a categorical variable (early vs. late) instead of as a covariate. A Box–Cox power transformation was required for the dependent variable ‘A260/230 ratio’ in order to meet the assumption for a linear model—‘normality of the residuals’. First, a full model, including all interaction terms, was fitted and based on the nonsignificant results; a simpler model with only the main effect could be retained. Tukey’s post hoc test [30] was performed to compare the mean differences between different levels of independent variables. Statistical analysis was performed using statistical packages implemented in R [31] and Jamovi [32]. The statistical test for significant difference was complemented with a test of equivalence. The test for equivalence, using TOST (Two One-Sided Test) [33], was performed using the ‘TOSTER’ module available in the Jamovi statistical software. The effect size parameter, Cohen’s d, was used to indicate the standardised difference between two means. For the equivalence test, a value of 0.5 for the effect size parameter was chosen, assuming moderate effect between euthanasia methods, storage condition and tissue types. This implies that if the means of two groups do differ by 0.5 standard deviations or more, even an insignificant effect cannot be considered equivalent.

## 3. Results

### 3.1. Descriptive Statistics

Initially, the set of samples with RNA measurements (N = 489) was analysed. The ANOXIA^TM^ group had a higher mean RNA concentration (604.56 ng/µL, 462.61 SD) and RIN values (8.85, 1.00 SD) compared to the T-61^®^ group (525.45 ng/µL, 404.30 SD and RIN value (8.77, 1.32 SD). Both the quality parameters, A260/230 and A260/280 ratios, had comparable measurements between the two groups. Similarly, on comparing the two storage conditions, samples stored in RNA*later*^TM^ had a higher average RNA concentration than the ones stored in LN_2_. Moreover, the A260/230 ratios for the samples stored in RNA*later*^TM^ had substantially higher values compared to the samples stored in LN_2_. Moreover, the variation among the quality parameters was generally lower with RNA*later*^TM^ storage compared to storage in LN_2_ (refer to Table 1).

The tissue from kidney, liver, pituitary and testis had higher average RNA concentration compared to that from CM, heart, hypothalamus, lungs, DIR and SIR (Figure 1A). Detailed descriptive statistics on the experimental data are given in Appendix A Appendix A. Comparison of RNA concentrations and quality parameters measured on the ten different tissue types are represented in Figure 1A,B and Appendix A. 

### 3.2. Data Filtering Based on Thresholds Set for RNA Measurements 

The filtering step removed 63 samples (see Appendix A Appendix A). Cremaster muscle was the major tissue type (*n* = 25) that was discarded due to poor RNA measurements, followed by inguinal rings DIR and SIR and heart (N = 9 each). Detailed descriptive data on the retained samples are given in Appendix A Appendix A. Out of the discarded samples, 36 samples belonged to the nitrogen anoxia group, and 27 samples were from the T-61^®^ group. 

Details of discarded samples belonging to the two euthanasia groups and storage conditions are plotted in Figure 2. A total of 426 samples (87%) were retained for further analysis. Details of retained samples, classified based on tissue types are plotted in Figure 3.

### 3.3. Linear Model to Determine Factors Influencing Dependent Variables

#### 3.3.1. RNA Concentration

Upon fitting the full model, the interaction effects were nonsignificant, and hence, a simple model with only the main effects was fitted. The final model was as follows:RNA concentration ~ Euthanasia Method+Storage+Tissue Type+Tissue Weight+Time of Sampling

The results from the linear model analysis and ANOVA are given in Table 2. Based on the analysis, euthanasia method, tissue type and tissue weight significantly influenced the RNA concentration (*p* < 0.001), whereas no significant effect of the storage condition was observed on the response variable (*p* = 0.49). The results indicated that the sampling time did not have a significant effect on the RNA concentration. Post hoc analysis results arrived at a significantly higher RNA concentration for tissues representing the nitrogen anoxia group (761.7, SE = 39.5) compared against the T-61^®^ group (647.7, SE = 42.9). Similarly, significantly higher values were associated with samples representing the tissue from testis, kidney and liver. The results from the Tukey’s test are plotted in Appendix A Appendix A. 

#### 3.3.2. RNA Quality—A260/230 Ratio

Since the residuals did not follow a normal distribution, a transformation to the power 4 was made on the A260/230 ratio before carrying out the full model analysis (see Appendix A Appendix A for a histogram before and after transformation). A simple model was later arrived at, after taking nonsignificant interaction terms out of the equation. The final model was as follows:(A260/230 ratio)^4  ~ Euthanasia Method+Storage+Tissue Type +Time of Sampling

Based on the analysis, it was found that the tissue type significantly influenced the RNA quality parameter A260/230 ratio (*p* < 0.001). However, no significant effect (*p* > 0.1) of euthanasia method and storage condition was observed on the response variable. Sampling time did not have an influence on the response variable, and in addition, none of the interaction terms in the model was significant. These results were also cross checked visually by looking at the relationships between the variables using pairs plot (Appendix A Appendix A).

Results from the linear model and ANOVA are given in Table 2. Since the nontransformed data also gave similar results during the regression analysis, post hoc analysis was performed using the nontransformed data to avoid complications associated with back transformations of the data. The post hoc analysis results indicated that the samples representing the tissue from pituitary and kidney had significantly higher A260/230 ratios compared to other tissue types. The results from the Tukey’s test comparing means of A260/230 ratios for the different tissue types are shown in Figure 4. 

#### 3.3.3. RNA Quality—A260/280 Ratio

The other RNA quality parameter, A260/280 ratio, was used to evaluate the euthanasia method and storage condition based on the following regression model: (A260/280 ratio) ~ Euthanasia Method+Storage+Tissue Type+Time of Sampling

The results from the analysis indicated that this quality parameter was significantly affected by the type of euthanasia method used as well as the tissue type. However, the difference in estimates was not technically relevant as pure RNA has a A260/280 ratio that ranges between 2.0 and 2.1 and all estimates fell within this range. The results from the regression analysis are given in Table 2.

#### 3.3.4. RNA Integrity Number (RIN)

RIN values were measured for a subset of samples that were grouped based on euthanasia method and storage conditions. The following regression model was fitted for evaluating the effect of these predictor variables on RIN value.
RIN  ~ Euthanasia Method+Storage+Tissue Type+Time of Sampling

Results indicated that the storage condition and tissue type had significant effect on the integrity of RNA (*p* < 0.001). However, euthanasia method and time of sampling did not have any influence on the estimated RIN value. Upon comparison of the mean RIN values, samples stored in RNA*later*^TM^ had a significantly higher value (9.25, SE = 0.11) compared to LN_2_ (7.99, SE = 0.31). Results from the post hoc analysis comparing the tissue types also confirmed the differences in RIN values among the different tissue types (Appendix A Appendix A). 

### 3.4. Test of Equivalence

An equivalence test was used to test whether the proof of no difference (nonsignificant) could be interpreted as a proof of equivalence between the predictor variables. Previously, we showed that euthanasia method did not significantly affect the quality parameter A260/230 ratio and RIN value. Therefore, an equivalence test was performed using the A260/230 ratio and RIN value as the response variables and euthanasia method as the grouping variable. The results from the equivalence test (Figure 5) show that the 90% confidence interval of the means fell within the range of upper bound and lower bound TOST interval for the A260/230 ratio. The significant t-test (*p* < 0.001) further confirms the equivalence between ANOXIA^TM^ and T-61^®^. However, the 90% confidence interval for the mean RIN values fell slightly outside the range of upper bound, together with a nonsignificant t-test (*p* > 0.07) for the upper bound TOST interval. The results from the TOST are given in Table 3.

The hypothesis of equivalence between the two euthanasia methods ANOXIA^TM^ and T-61^®^ is tested using the two one-sided ‘*t*’ tests on A260/230 ratio and RIN values. The greatest of the two *p* values is taken as the *p* value of the equivalence test. Since *p* value for the upper bound of the TOST interval is nonsignificant, an equivalence between the euthanasia methods cannot be proved based on RIN values. 

## 4. Discussion

The present study examined the suitability of a euthanasia method that used an inhalant anaesthetic ‘nitrogen gas’ in foam in an anoxia box (ANOXIA^TM^) for euthanasia of piglets used in research. The anoxia method was compared with the T-61^®^ injection method for euthanizing the animals. The yield and quality of RNA extracted from various tissue types were evaluated for this purpose. Since these samples were stored in two different conditions (RNA*later*™ and LN_2_), their interaction effects with the euthanasia methods were also examined.

In this study, no significant difference was found in A260/230 ratios and RIN values between the ANOXIA^TM^ and T-61^®^ euthanasia methods in male piglets. However, the yield of RNA was significantly higher for the anoxia method than that observed in the case of the T-61^®^ method. For the above reasons, the nitrogen anoxia method might be considered as a suitable alternative for euthanasia of piglets used in research. 

Although there was proof for no difference between the two euthanasia methods based on the quality parameters, an equivalence between the two euthanasia methods could only be proven with regard to the A260/230 ratio and not with the RIN value. Additionally, any interaction effect of the storage conditions and the euthanasia methods was found to be insignificant in our study. The significant effect of euthanasia methods on the A260/280 ratio was technically irrelevant as the respective effect sizes for this parameter was within the accepted range for pure RNA.

We observed that different tissue types showed varying levels of RNA yield, which can be attributed to be due to their different metabolic activity levels. Liver, kidney, brain and testis have been reported to have a high metabolic rate compared to skeletal muscle and adipose tissue [34,35]. In the current study, it was observed that liver, kidney, testis and pituitary tissue had significantly higher amount of RNA as compared against other tissue types. Tissue weights also significantly affected the total RNA yield, specifically for these metabolically active tissue types. Age and sex could influence the gene expression levels in different tissue types [36,37], but it is not certain whether these factors would affect the total RNA yield and quality parameters. However, the validity of our study should be further confirmed because our sample size was low (*n* = 12) and it was conducted only in young male animals.

Time of sampling did not have any influence on the RNA quantity and quality parameters compared in this study. This was in contrast to the previously reported results [2,3,4] where a delay in the sample processing negatively impacted RNA quantity and quality. The possible explanation for this disparity is that the early and late samples differed only by an approximate time difference of 30–40 min and the tissue types compared in this study mainly constituted the metabolically active tissues. 

Advantages of using RNA*later*™ for tissue storage purposes, especially in field settings, have been already mentioned in previous studies [23,24]. The results of this study revealed that the integrity of RNA (RIN) for tissues stored in RNA*later*™ was significantly better than that for the samples stored in LN2. The variation among the quality parameters was generally lower with samples stored in RNA*later*™ compared to storage in LN2. Furthermore, more samples that had to be discarded based on quality thresholds were the ones stored in LN2. This could also explain why a significant effect of the storage condition was not detected in our analysis with the filtered data but with the prefiltered data. Time of sampling was found to have no influence on the estimated RIN value. However, it is to be noted that there were no ‘late’ samples included in this analysis, and the average sampling times was around 15 min.

The importance of RNA purity is widely acknowledged, especially for RT-qPCR analyses [12]. RNA concentration will be overestimated if the samples are low in purity and thus results in unreliable outcomes from the analysis. Similarly, the library preparation step for RNAseq analysis could also be affected due to the overestimation of RNA concentration. Here, it will be difficult to decide on the number of PCR cycles or on the input volume required during the library preparation step. Hence, additional purification and clean-up steps are required before proceeding further with downstream analysis [22,38]. 

The importance of RNA*later*™ for storing tissues for RNA studies gains more support from the above reasoning. However, there are recent publications discussing unfavourable effects of RNA*later*™ on gene expression profiles [39,40]. Upon RT-qPCR analysis, differential levels of gene expression were observed in lung, heart, liver and skeletal muscle tissues from albino rats stored in RNA*later*™ compared to snap freezing. Similarly, storage of Arabidopsis seedlings in RNA*later*™was reported to have resulted in altered transcriptomes as a response to osmotic stress caused by the ammonium salts present in the storage media [39]. This warrants the need for careful considerations and further research regarding the choice of storage medium, especially for differential gene expression studies. This also reminds us about the importance of future research focused on the transcriptome level to know the impact of the euthanasia methods and storage conditions on gene expressions. 

The findings from our research could not specify any reasons against the suitability of the nitrogen anoxia method for euthanasia of piglets used in research. This euthanasia method has advantages of operational ease and less animal handling. If future research can confirm the claim of improved animal welfare while using the nitrogen anoxia method, as discussed by Lindahl et al. [6], this can be chosen as a suitable alternative euthanasia method. However, there were limitations for this present study due to the small number of animals tested and also a possible bias because only male animals were used. Further research should focus on confirming the suitability of this euthanasia method for transcriptomics studies by extending this research to gene expression levels.

## 5. Conclusions

The present study could not find any difference between the nitrogen anoxia and T-61^®^ euthanasia methods in male piglets based on the RNA quality parameters. However, the yield of RNA was significantly higher for the anoxia method than that observed in the case of the T-61^®^ method. The nitrogen anoxia-based euthanasia method could be a suitable replacement for other euthanasia techniques, but the claims of improved animal welfare must be further validated before making this conclusion.

## Figures and Tables

**Figure 1 animals-13-00698-f001:**
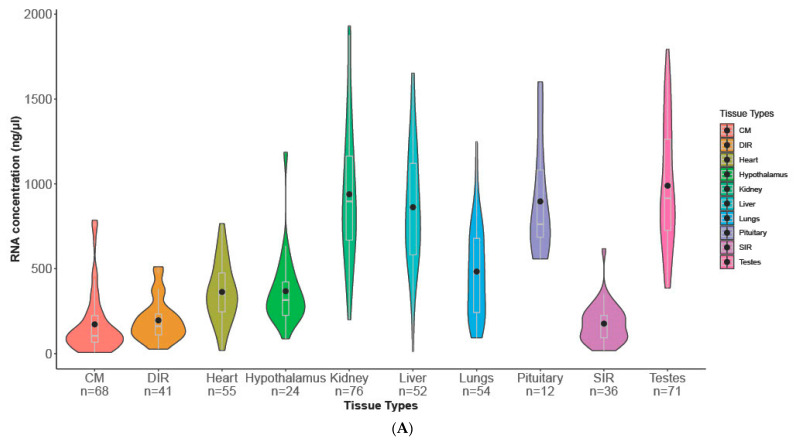
(**A**) RNA concentrations from the ten different tissue types are plotted. Kidney, liver, pituitary and testis have higher average RNA concentrations in comparison with cremaster muscle, inguinal rings, heart and hypothalamus. CM: cremaster muscle; DIR: deep inguinal ring; SIR: superficial inguinal ring. (**B**) RNA quality estimated using the A260/230 ratio from the ten different tissue types are plotted. Cremaster muscle had the lowest average value with a mean of less than 1.5, followed by inguinal rings (DIR and SIR). CM: cremaster muscle; DIR: deep inguinal ring; SIR: superficial inguinal ring.

**Figure 2 animals-13-00698-f002:**
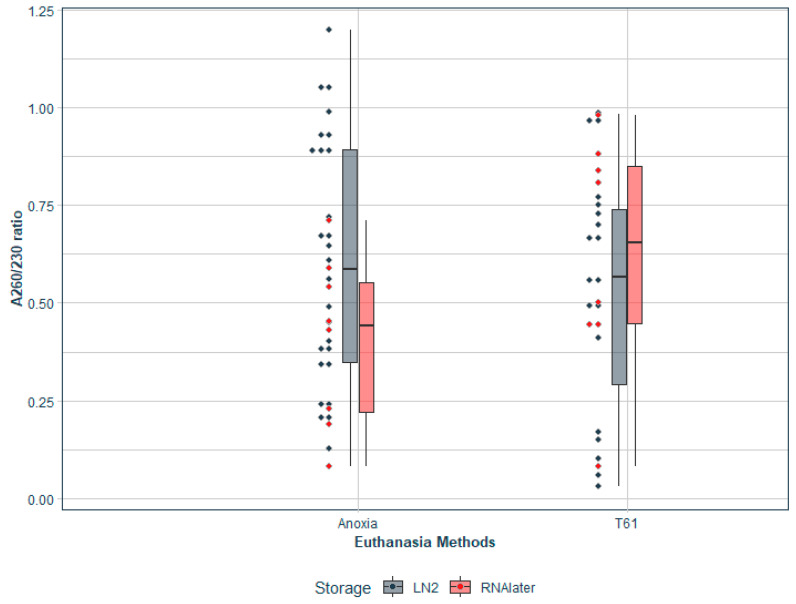
Details of discarded samples grouped based on euthanasia method and storage conditions is plotted here. Black dots represent the samples stored in LN_2_, and red dots represent the samples stored in RNA*later*^TM^. Samples stored in LN_2_ (*n* = 47) were the majority (~75%) of samples that were discarded compared to RNA*later*^TM^ storage (*n* = 16).

**Figure 3 animals-13-00698-f003:**
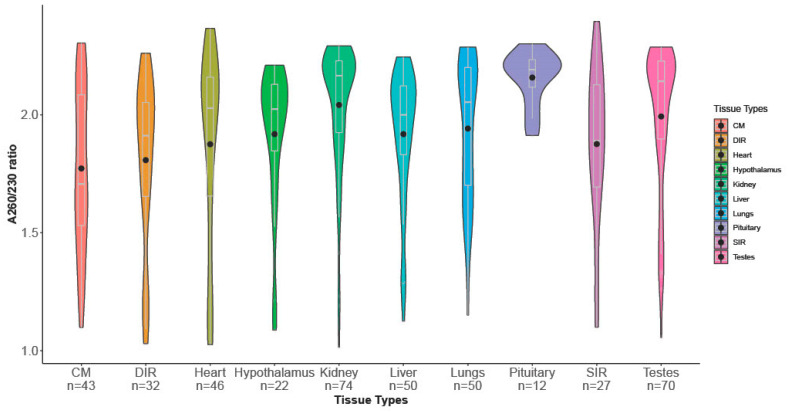
Details of samples retained after filtering. All samples with RNA concentration above 50 ng/µL and a A260/230 ratio above 1.0 were retained for further evaluation. CM: cremaster muscle; DIR: deep inguinal ring; SIR: superficial inguinal ring.

**Figure 4 animals-13-00698-f004:**
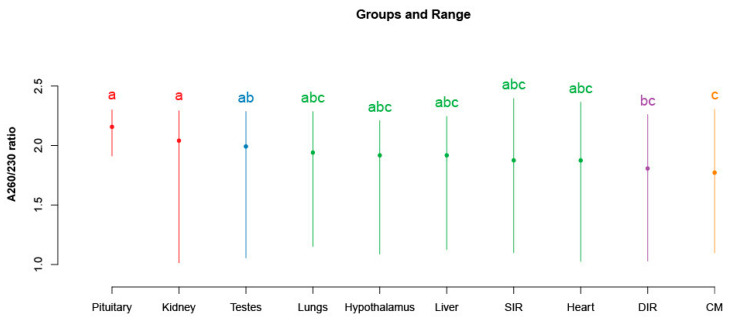
Post hoc analysis confirms the significant effect of tissue types on the A260/230 ratio. Range of the A260/230 ratio and means (dots) per tissue types are marked in the figure. Groups carrying different superscript letters and colours differ significantly (*p* < 0.05) CM: cremaster muscle; DIR: deep inguinal ring; SIR: superficial inguinal ring.

**Figure 5 animals-13-00698-f005:**
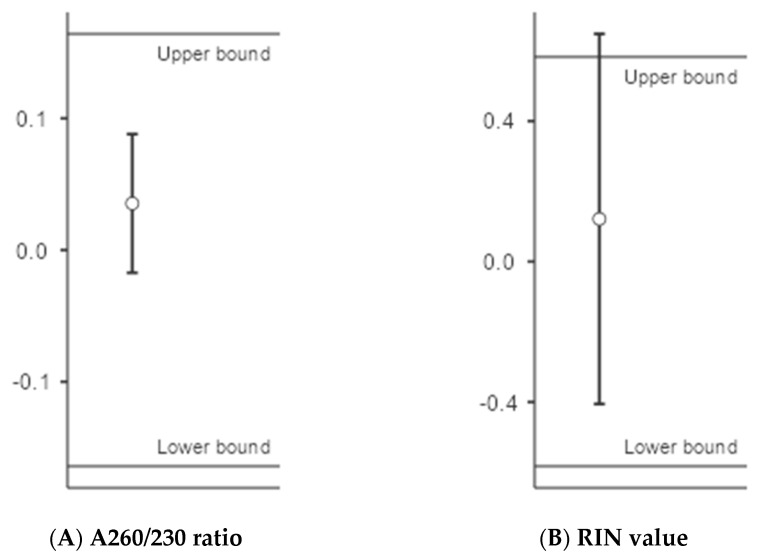
Equivalence plotted using confidence interval approach to explain equivalence between euthanasia methods (ANOXIA^TM^ vs. T-61^®^) on the variables (**A**,**B**). The vertical bars depict the upper and lower limits for the effect size parameter, Cohens d. RIN values fall beyond the confidence interval at the upper bound limit, and hence, the two euthanasia methods cannot be considered equivalent based on the RIN values.

**Table 1 animals-13-00698-t001:** Mean and standard deviations of RNA quality parameters for the experimental samples classified based on euthanasia method and storage condition.

	RNA Concentration (SD)	A260/230 Ratio (SD)	A260/280 Ratio (SD)	RIN (SD)
ANOXIA^TM^	604.56 (462.61)	1.77 (0.57)	2.13 (0.04)	8.85 (1.00)
T-61^®^	525.45 (404.30)	1.75 (0.56)	2.11 (0.04)	8.77 (1.32)
RNA*later*^TM^	611.99 (433.88)	1.86 (0.47)	2.12 (0.03)	9.26 (0.66)
LN2	525.12 (439.57)	1.66 (0.63)	2.12 (0.05)	7.99 (1.40)

**Table 2 animals-13-00698-t002:** Modelling results for RNA concentration, A260/230 ratio, A260/280 ratio and RIN value as dependent variables: F-test, ls-means and estimates obtained in a linear model.

	RNA Concentration	A260/230 Ratio	A260/280 Ratio	RIN
	Final Model	Final Model	Final Model	Final Model
**N**	239	426	426	55
**Overall mean**	633.3	1.93	2.116	8.79
**RMSE**	248.18	0.31	0.02	0.63
** *p* ** **values of F-test**				
**FACTORS**				
Euthanasia Method	***	0.25	***	0.51
Storage	0.49	0.18	0.47	***
Tissue Type	***	***	***	***
Time of Sampling	0.41	0.18	*	0.56
Tissue Weight	***			
**LSmeans for treatment levels (std. error in brackets)**				
**Euthanasia Method**				
ANOXIA^TM^	761.7 (39.5) ^a^	1.95 (0.02) ^a^	2.12 (0.002) ^a^	8.85 (0.18) ^a^
T-61^®^	647.7 (42.9) ^b^	1.91 (0.02) ^a^	2.11 (0.002) ^b^	8.73 (0.26) ^a^
**Storage Condition**				
RNAlater	729.4(40.0) ^a^	1.95 (0.02) ^a^	2.117 (0.002) ^a^	9.25 (0.11) ^a^
LN_2_	703.7(44.0) ^a^	1.90 (0.03) ^a^	2.114 (0.002) ^a^	7.99 (0.31) ^b^
**Estimates for Time of Sampling and Tissue Weight (std. error in brackets)**				
**Time of Sampling (minutes)**	0.94 (0.72)	−0.001 (0.001)	0.13 (0.05)	0.04 (0.07)
**Tissue Weight** **(milligram)**	21.02 (3.37)			

Significant *p* values from the F-test are represented as * *p* < 0.05, *** *p* < 0.001. Groups carrying different superscripts differ significantly (*p* < 0.05).

**Table 3 animals-13-00698-t003:** Summary of the Test on Independent Samples *t*-test—Two One-Sided Test Results.

			t	df	*p*
**A260/230** **Ratio**	*t*-test		1.109	424	0.268
	TOST Upper		−4.03	424	<0.001
	TOST Lower		6.25	424	<0.001
**RIN**	*t*-test		0.385	53	0.701
	TOST Upper		−1.47	53	0.074
	TOST Lower		2.24	53	0.015
**Equivalence Bounds**					
				90% Confidence interval	
		Low	High	Lower	Upper
**A260/230** **Ratio**	Cohen’s d	−0.5	0.5		
	Raw	−0.164	0.164	−0.0172	0.0881
**RIN**	Cohen’s d	−0.5	0.5		
	Raw	−0.583	0.583	−0.4057	0.6484

TOST Upper—Upper limit of confidence interval; TOST Lower—lower limit of the confidence interval.

## Data Availability

The data presented in this study are available on request from the corresponding author.

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
