# Peer review of "A Comparative Study on the Effect of Euthanasia Methods and Sample Storage Conditions on RNA Yield and Quality in Porcine Tissues"

_animals, 2023, doi:10.3390/ani13040698_

Round 1
Reviewer 1 Report
Overall, the manuscript is well written and report a technically sound research project.
I only have some minor points that, in my opinion, require attention:
- the authors report the time elapsed between the beginning our euthanasia and tissue sampling, but it would have been interesting to see the time passed since the declaration of death;
- how was death confirmed (e.g. EKG, Auscultation etc..)?
- the authors administered T-61® upon intra-cardiac injection. Since heart was sampled, IV may have been a safer choice (auricular veins/femural artery can be easily accessed in 1 week old piglets). I think it would be important to mention in the discussion that the heart was punctured with potential bias to the study (in all honesty, I do not think this is that big of an issue, but it is worth mentioning).
- some figures have poor definition (see fig.5), please fix it.
- tables should be formatted to better fit the template.
Author Response
Dear Sir/Madam,
We would like to thank you for reviewing our manuscript and for your valuable comments. We have gone through your suggestions and have made these relevant changes. Revised portion are marked as track changes in the paper. The response to your comments are in the attached documents. Please verify the same.
Kind regards,

Reviewer 2 Report
After reviewing the paper entitled “A comparative study on the effect of euthanasia methods and sample storage conditions on RNA yield and quality in porcine tissues” it is worth mentioning that the manuscript is relevant and provides knowledge that the euthanasia method using nitrogen anoxia technique or T-61 not affect RNA quality parameters while storage in RNA later significantly increases the RNA integrity in comparison to snap freezing different tissue types. Any interaction effect of the storage conditions and the euthanasia methods was found. The results are well documented and discussed and the paper provides knowledge about the effect of euthanasia methods and sample storage conditions on RNA yield and quality in porcine tissue. Respecting that pigs are well research model used also in biomedical study the presented results and knowledge will be helpful for many scientists involved in research with porcine tissues.
However, it should be highlighted that samples were collected from one week old purebred male piglets. The question arises whether post-pubertal pigs tissues would response similarly as observed in the study. Moreover, the reason for selection only one week old male piglets should be provided.
Respecting the konwledge on the meaning of tissue metabolic level and/or the effect of hormonal status of animal, consider how the sex and/or the age and puberty may affect studied parameters and enhance the discussion section including these aspects, please. Moreover, please precise the title includindg information that the study were ptovided on male pre-puberal piglets. In my opinion it is a very importatnt issue. Thank you.
Author Response

(The authors gave the same response as above.)
